# c-Kit Induces Migration of Triple-Negative Breast Cancer Cells and Is a Promising Target for Tyrosine Kinase Inhibitor Treatment

**DOI:** 10.3390/ijms23158702

**Published:** 2022-08-05

**Authors:** José A. López-Mejía, Luis F. Tallabs-Utrilla, Pablo Salazar-Sojo, Jessica C. Mantilla-Ollarves, Manuel A. Sánchez-Carballido, Leticia Rocha-Zavaleta

**Affiliations:** 1Departamento de Biología Molecular y Biotecnología, Instituto de Investigaciones Biomédicas, Universidad Nacional Autónoma de México, Ciudad de Mexico 04510, Mexico; 2Escuela de Medicina, Universidad Panamericana, Ciudad de Mexico 03920, Mexico; 3Programa Institucional de Cáncer de Mama, Instituto de Investigaciones Biomédicas, Universidad Nacional Autónoma de México, Ciudad de Mexico 04510, Mexico

**Keywords:** triple-negative breast cancer, c-Kit, receptor tyrosine kinase, tyrosine kinase inhibitor, stem cell factor, nilotinib

## Abstract

Triple-negative breast cancer (TNBC) is associated with a poor prognosis and the absence of targeted therapy. c-Kit, a receptor tyrosine kinase (RTK), is considered a molecular target for anticancer drugs. Tyrosine kinase inhibitors (TKIs) recognizing c-Kit are used for the treatment of c-Kit-expressing tumors. However, the expression, function, and therapeutic potential of c-Kit have been little explored in TNBC. Here, we studied the expression and effects of c-Kit in TNBC through in vitro and in silico analysis, and evaluated the response to TKIs targeting c-Kit. Analysis of TNBC cells showed the expression of functional c-Kit at the cell membrane. The stimulation of c-Kit with its ligand induced the activation of STAT3, Akt, and ERK1/2, increasing cell migration, but had no effect on cell proliferation or response to Doxorubicin. Analysis of public datasets showed that the expression of c-Kit in tumors was not associated with patient survival. Finally, TNBC cells were susceptible to TKIs, in particular the effect of Nilotinib was stronger than Doxorubicin in all cell lines. In conclusion, TNBC cells express functional c-Kit, which is a targetable molecule, and show a strong response to Nilotinib that may be considered a candidate drug for the treatment of TNBC.

## 1. Introduction

According to estimates that were produced by the International Agency for Research on Cancer, breast cancer is the neoplasm with the most incidents in women worldwide [1]. In clinical settings, breast tumors are classified by the presence of estrogen receptors (ER), progesterone receptors (PR), and human epidermal growth factor receptor 2 (HER2) amplification. Tumors lacking hormone receptors and HER2 are defined as triple-negative breast cancer (TNBC), the prevalence of TNBC varies across countries from 6.8% of all breast cancers in The Netherlands to 27.9% in India [2]. The development of targeted therapies for ER/PR-positive and HER2-overexpressing tumors has improved the clinical outcome of patients, whereas women that are diagnosed with TNBC remain the group of breast cancer patients with the poorest prognosis, due to the absence of molecular targets and the consequent limitation of treatment options.

In recent years, attempts that are dedicated to identifying molecular markers in TNBC have demonstrated that this type of breast cancer is remarkably heterogeneous [3]. Analysis of public expression datasets suggested that TNBC can be divided into discrete subtypes that are associated with different prognoses and responses to chemotherapy [3,4,5]. An examination of genomic and transcriptomic profiling of TNBC samples has allowed the identification of targetable molecules in TNBC subtypes. Among them, receptor tyrosine kinases (RTKs) such as c-Kit were found to be significantly overexpressed in discovery and validation sets [3,5]. The activation of RTKs and their associated signal transduction cascades regulates cellular proliferation, differentiation, and motility. Thus, the dysregulation of RTKs, particularly c-Kit, has been linked to tumor progression [6]. Several studies have demonstrated the role of c-Kit in gastrointestinal stromal tumors, thymic carcinoma, melanoma, leukemias, and even mastocytosis [7]. However, the specific functions and potential usefulness of c-Kit as a target molecule in TNBC are not completely understood.

c-Kit is the tyrosine kinase receptor for stem cell factor (SCF). c-Kit was originally characterized as a regulator of hematopoiesis [8] and melanogenesis [9]. Nevertheless, it is now recognized as a proto-oncogene due to its association with different types of cancer. Binding of SCF induces the formation of c-Kit dimers that stimulates homologous transphosphorylation of the receptor. Phosphorylated c-Kit promotes the activation of multiple downstream signaling cascades, such as: the phosphatidylinositol 3-kinase (PI3K)/protein kinase B (Akt) that induces cell proliferation; the mitogen-activated protein kinase (MAPK) that functions as a survival signal [10]; and the Janus kinase (JAK)/signal transducer and activator of transcription (STAT) transduction pathway that has been associated with cell proliferation and response to cytotoxic agents [11]. The expression of c-Kit has been demonstrated in TNBC [12], but the effect of c-Kit activation by SCF on signal transduction and cellular responses has not been fully explored.

Several selective small molecules have been designed to target and inhibit c-Kit. Imatinib was the first inhibitor that was approved by the Food and Drug Administration (FDA) to treat c-Kit-expressing gastrointestinal stromal tumors. Later, new tyrosine kinase inhibitors (TKIs) were developed, some of them can block c-Kit function, i.e., Dasatinib, Sorafenib, Sunitinib, and Nilotinib, and they are commercially available. Currently, Lapatinib is the sole TKI-approved to treat breast cancer, but it is exclusively indicated for patients with HER2-positive breast cancer because it targets epidermal growth factor receptors [13]. Unfortunately, research on the potential cytotoxic effect of TKIs targeting c-Kit in TNBC has been hampered by the limited understanding of the effects of c-Kit activation in TNBC cells. For this reason, in the present work we investigated the expression of c-Kit in TNBC-derived cells lines; evaluated the effect of SCF on the activation of signaling cascades, cell proliferation, and migration; and the response to standard chemotherapy. We also integrated public genomic and expression data to elucidate the expression and potential impact of c-Kit and SCF in TNBC patients. In addition, we evaluated the effect of commercially available TKIs targeting c-Kit and compared their cytotoxic efficiency with that of Doxorubicin in TNBC cells.

## 2. Results

### 2.1. TNBC Cells Express Different Levels of Membrane c-Kit

The expression of c-Kit was studied in four TNBC-derived cell lines. An analysis of gene expression showed the presence of c-Kit transcripts of the same size to those that were observed in the control K562 cells (Figure 1a). The presence of the c-Kit protein was also confirmed by Western blot (Figure 1b). To be functional, c-Kit must be located at the cell surface, thus we analyzed the presence of the receptor by flow cytometry. As seen in Figure 1c, membrane c-Kit was detected in TNBC cells; the proportion of cells showing surface c-Kit varied from 43.17% in HCC-1806 to 88.92% in HCC-70. It is known that gain of function mutations are frequently detected in c-Kit that is expressed in human tumors. Most oncogenic mutations are located within exons 9, 10, 11, 13, and 17, which encode the kinase domain and the transmembrane region of the receptor [14]. Thus, to know whether TNBC cells have mutations that are associated with the constitutive activation of c-Kit, we sequenced exons 9, 10, 11, 13, and 17. In contrast with other tumors we did not find any change in the sequences that were analyzed, suggesting that the cells that were included in this work display a c-Kit with a normal function.

Our results showed that c-Kit is expressed in TNBC-derived cell lines. To further evaluate the expression of c-Kit in tumor samples we examined public gene expression datasets. As we demonstrated that the presence of c-Kit mRNA in all cell lines showed a good relationship with the level of c-Kit protein, we considered that using c-Kit gene expression as a surrogate measurement of protein was acceptable. There were two datasets that were explored: METABRIC (Molecular Taxonomy of Breast Cancer International Consortium) comprising of DNA and RNA sequencing data from 299 primary TNBC tumors [15,16], and The Cancer Genome Atlas project (TCGA) [17] including data from 168 TNBC tumors. Tumors showing expression values that were two times higher than the mean were assumed to have c-Kit overexpression. As shown in Figure 1d, overexpression of c-Kit was detected in 11% (33/299 samples) and 8% (13/168 samples) in the METABRIC and TCGA cohorts, respectively. In addition, the sequencing data demonstrated the absence of mutations in the c-Kit gene, which is in agreement with what was observed in the cell lines. Furthermore, we evaluated the expression of the c-Kit ligand SCF in human samples, observing evidence of SCF overexpression in 0.7% (2/299 samples) and 0% (0/168 samples) in the METABRIC and TCGA cohorts, respectively. No mutations in the SCF gene were reported.

### 2.2. SCF Induces Activation of c-Kit in TNBC Cells and Promotes Cell Migration

Our results indicated that TNBC cells express c-Kit at the cell membrane. To determine whether the receptor is functional, the cells were incubated with increasing concentrations of SCF and the receptor transphosphorylation was evaluated. As seen in Figure 2a, SCF stimulation induced c-Kit phosphorylation from 1 min. These observations confirm that TNBC-expressed c-Kit can be activated by its natural ligand. The activation of c-Kit has been demonstrated to modify cell proliferation in various cancer cells [18,19]; thus, we next evaluated the potential effect of c-Kit activation on TNBC cell proliferation. A time-course study exposing the cells to increasing concentrations of SCF was set, and the results revealed that c-Kit activation did not increase the proliferation of TNBC cells at 48 h (Figure 2b) or longer incubation time points (Appendix A). We have previously demonstrated that c-Kit activation stimulates motility of cervical cancer cells [20]. Consequently, to determine whether c-Kit regulates cell migration, TNBC cells were incubated with 1, 10, and 100 ng/mL SCF and tested in a Boyden Chamber-based migration assay, the results are shown in Figure 2c. A significant increment of migrating cells was observed when the HCC-1806 were stimulated with 100 ng/mL SCF (*p* < 0.05), whereas HCC-1937 cell motility was significantly increased by 1, 10 ng/mL (*p* < 0.05), and 100 ng/mL SCF (*p* < 0.005). In sharp contrast, migration of HCC-70 and MDA-MB-468 was not stimulated by SCF (Figure 2c). In order to demonstrate that cell motility was caused by the activation of c-Kit, the cells were pre-incubated with 10 µM ISCK03, which inhibits c-Kit activation. Pre-incubation of both HCC-1806 and HCC-1937 cells with the ISCK03 inhibitor completely abolished the capacity of SCF to promote cell motility (Figure 2d), suggesting that the effect is mediated by the activation of c-Kit.

### 2.3. SCF Induces Activation of STAT3, Akt and ERK1/2

Our results indicated that the activation of c-Kit by SCF promotes the migration of HCC-1806 and HCC-1937 cells. It has been reported that SCF-induced cell motility is mediated by the activation of the MAPKs [21], the PI3K/Akt [22], and the JAK2/STAT3 [23] pathways. Thus, to determine if c-Kit triggers migration signaling cascades, the cells were incubated with SCF and the time-course phosphorylation of clue molecules was documented. We first analyzed the phosphorylation of the kinases ERK1/2, which are downstream effectors of the MAPKs cascade. The evaluation of ERK1/2 showed a constitutive and sustained phosphorylation in HCC-1806, HCC-1937, and MDA-MB-468. In contrast, in HCC-70 cells, SCF induced the activation of ERK1/2 after 1 min stimulation. Interestingly, in all the cell lines the level of ERK1/2 phosphorylation fluctuated during the time course, but remained phosphorylated until the end of the assay (Figure 3). On the other hand, the assay showed that Akt was activated by SCF in HCC-1806, HCC-1937, and HCC-70 cells. In all cases the phosphorylation of Akt was clearly observed from 1 min after stimulation. Unlike the other cell lines, the MDA-MB-468 cells showed a highly phosphorylated Akt even before the stimulation with SCF, and the level of Akt activation did not increase with time (Figure 3). We finally investigated the activation of STAT3, a crucial downstream effector of the JAK/STAT signaling pathway. The transcription factor STAT3 was activated by SCF in all the cell lines. The highest level of STAT3 phosphorylation was reached at 1 min after stimulation in HCC-1806 and HCC-70, 30 min in MDA-MB-468, and 15 min in HCC-1937 (Figure 3). In all cases, STAT3 remained phosphorylated until the end of the experiment. These observations suggest that the PI3K/Akt and JAK/STAT3 signaling cascades are readily activated by c-Kit in cells with the capacity to migrate as a response to SCF stimulation.

### 2.4. The c-Kit/SCF Axis Has No Effect on TNBC Cell Survival

It is known that STAT3 plays a major role as a regulator of resistance to chemotherapy in breast cancer [24]. Since we observed that SCF induced the activation of STAT3 in TNBC cell lines, we wanted to explore whether the activation of c-Kit hinders the response of TNBC cells to Doxorubicin, which is frequently used to treat patients with TNBC. The cells were pre-incubated with increasing concentrations of SCF and then exposed to Doxorubicin. Cell viability was evaluated after 48 h; the results are depicted in Figure 4a. The assay showed that none of the concentrations of SCF were able to hamper the cytotoxic effect of Doxorubicin. These results suggest that the activation of c-Kit has no impact on TNBC cell survival. Thus, to extend our observations, we determined whether the level of c-Kit expression is associated with overall survival and relapse-free survival of TNBC patients. According to data from the METABRIC database, patients with c-Kit overexpression showed a trend of having reduced 5-year overall survival compared to the unaltered group, but the difference was not significant (*p* = 0.566) (Figure 4b). Accordingly, at 10-years follow-up, both groups showed a similar level of overall survival (*p* = 0.566) (Figure 4c). Likewise, analysis of data demonstrated that the group with c-Kit overexpression and the unaltered group had a similar level of 5-year relapse-free survival (Figure 4d). Taken together, these results seem to suggest that the expression of c-Kit is not relevant for tumor cell survival.

### 2.5. Responses of TNBC Cells to Tyrosine Kinases Inhibitors

RTKs such as c-Kit have been considered as target molecules for the development of small-molecule drugs known as tyrosine kinase inhibitors (TKIs). In fact, Imatinib, the first commercially available TKI, was designed to inhibit Bcr-Abl and c-Kit kinases. A number of TKIs targeting c-Kit has been further developed, so we studied their potential cytotoxic effect on c-Kit-expressing TNBC. To this end, we tested the response of TNBC cells to Imatinib, Dasatinib, Sorafenib, Sunitinib, and Nilotinib, which target c-Kit. It is important to mention that the selected TKIs are also able to target other RTKs and intracellular tyrosine kinases. The whole panel of molecules that are recognized by each TKI is shown in Appendix A. Lapatinib was included as a control because it does not target c-Kit (Appendix A). Doxorubicin was included in order to compare cytotoxicity that was induced by TKIs with that which was produced by traditional chemotherapy. The cytotoxic effect was expressed as the IC50 of each TKI. The results that are shown in Table 1 indicate that the TNBC cells were susceptible to all TKIs that were tested. However, distinct levels of response were detected. Imatinib had the weakest cytotoxic effect with calculated IC50 values >20,000 nM in all cell lines (51,800 nM in HCC-1806, 24,750 nM in HCC-1937, 24,800 nM in HCC-70, and 24,310 nM in MDA-MB-468). The response to Dasatinib was highly heterogenous; HCC-1806 were the most susceptible cells with an IC50 of 340 nM, followed by HCC-1937 (3120 nM), MDA-MB-468 (9210 nM), and HCC-70 (10,230 nM). The cytotoxic effect of Sorafenib also varied among cell lines. In this case, MDA-MB-468 were the most susceptible cells, with a calculated IC50 of 130 nM. The less susceptible cells were HCC-70 with an IC50 of 13,660 nM. The cytotoxic profile of Sunitinib was relatively high in the MDA-MB-468 cells (IC50 = 285 nM), compared with that which was observed in HCC-1806 (3640 nM), HCC-1937 (2910 nM), and HCC-70 (3420 nM). In sharp contrast, Nilotinib had the highest cytotoxic profile in all the cell lines, also showing the most homogenous response among the cell lines with IC50s < 50 nM (46 nM in HCC-1806, 11 nM in HCC-1937, 31 nM in HCC-70, and 23 nM in MDA-MB-468).

Since Doxorubicin is frequently used to treat TNBC, we compared the level of cytotoxicity that was mediated by Doxorubicin and TKIs. As shown in Table 1, the cytotoxic profile of Imatinib, Dasatinib, Sorafenib, and Sunitinb was weaker than the cytotoxicity that was mediated by Doxorubicin. Conversely, the level of cytotoxicity that was produced by Nilotinib was consistently higher than that of Doxorubicin. The IC50 of Doxorubicin was 3.9-fold higher than Nilotinib in HCC-1806, 17.3-fold higher in HCC-1937, 22.5-fold higher in HCC-70, and 5-fold higher in the MDA-MB-468 cells. These results indicate that Nilotinib has a remarkable cytotoxic capacity against TNBC cells.

Of note, although Lapatinib does not recognize c-Kit, it was able to kill HCC-1806 and HCC-70 cells more efficiently than Doxorubicin (IC50 26 vs. 180.5 nM, and 80 vs. 700 nM, respectively) (Table 1).

### 2.6. c-Kit Is a Dominant Targetable RTK in TNBC Tumors

Our observations suggest that c-Kit-expressing TNBC cells are susceptible to TKI cytotoxic activity. However, all TKIs that were tested are multi-targeted, and able to inhibit RTKs and intracellular tyrosine kinases, so we were interested in defining whether the RTK/tyrosine kinases expression profile was associated with the response of TNBC cells to TKIs. To this end, we first scrutinized data from the Gene expression-based Outcome for Breast cancer Online (GOBO) tool to determine the level of expression of target RTKs and intracellular tyrosine kinases. Only those target molecules showing a significantly higher level of expression in tumors of the basal subtype, which contains the highest number of TNBC samples, were selected to be evaluated in cell lines. The results showing the comparative level of RTKs and intracellular tyrosine kinases expression among breast cancer subtypes are shown in Appendix A. A total of seven intracellular tyrosine kinases (LCK, YES, FYN, FRK, LYN, BTK, ABL) and one RTK (c-Kit) were selected, and their expression was analyzed in TNBC cell lines by RT-PCR. As observed in Figure 5, different patterns of expression were observed in the cell lines. However, it was clear that the intracellular ABL, LYN, YES, and FRK tyrosine kinases were expressed in all the cell lines. This observation suggests that the effect of TKIs might be associated with the presence of various targets. It is worth mentioning that the expression of intracellular ABL may contribute to the strong effect of Nilotinib against TNBC cells.

To further investigate the expression of target intracellular tyrosine kinases and c-Kit in human tumors, we interrogated METABRIC transcriptomic datasets. In this platform, breast cancer patients are stratified according to the five intrinsic molecular subtypes using the PAM50 method. Thus, we restricted our analysis to patients of the basal subtype (N = 199), in which expression of HER2 was detected in only 12% samples. The proportion of patients showing overexpression (z-score relative to diploid samples > 2) and underexpression (z-score relative to diploid samples < −2) of each target molecule is depicted in Figure 6. As shown in Figure 6a, Dasatinib targets 15 tyrosine kinases, the more frequently overexpressed were LYN (18% samples), c-Kit (15% samples), and SRC (15% samples). Sorafenib targets 10 molecules, in this case kinases RAF, BRAF, and c-Kit were found to be overexpressed in 20%, 22%, and 15% samples, respectively (Figure 6b). From the eight kinases that were targeted by Sunitinb, c-Kit was the most frequently overexpressed in the population that was analyzed (15%) (Figure 6c). Similarly, c-Kit was more commonly overexpressed than the other five tyrosine kinases that were targeted by Imatinib (Figure 6d). Nilotinib only targets c-Kit and Abl1. c-Kit was overexpressed in 15% of the samples, and Abl1 was found overexpressed in 8% of the samples; none of the patients that were analyzed overexpressed both kinases (Figure 6e). Finally, Lapatinib has only two target molecules, EGFR and ERBB2 (HER2), as observed in Figure 6f, EGFR was overexpressed in 40% of the samples, whereas ERBB2 was found overexpressed in 12% of the patients. These data suggest that c-Kit is overexpressed in a proportion TNBC samples and might be considered a potential TKI target in a subgroup of TNBC patients.

## 3. Discussion

TNBC remains the type of breast cancer with the poorest prognosis. Efforts are currently focused on identifying molecular targets that might guide therapeutic strategies to improve clinical outcomes. c-Kit has been recognized as a tumor driver and targetable molecule in gastrointestinal stromal tumors [25], but its role in TNBC remains controversial. Here, we report that TNBC cell lines and tumors express c-Kit. Our analysis of transcriptomic datasets revealed that a small proportion of TNBC tumors (8–11%) overexpress c-Kit. However, immunohistochemistry analysis of TNBC samples has demonstrated the expression of c-Kit in a high proportion (30–89%) of tumors [26,27,28,29]. Moreover, Jansson and colleagues reported that positive c-Kit immunostaining was more frequently observed in TNBC (49%) than in non-TNBC patients (10%) [28]. In contrast, other authors have reported the loss of c-Kit expression during breast cancer development [30]. However, it is important to note that former analyses did not categorize samples by molecular or histological subtype, thus the expression of c-Kit in TNBC was not determined. Gain of function mutations have been identified as the main cause of c-Kit dysregulation in cancer. Despite that, we were unable to detect c-Kit mutations in cell lines. Likewise, c-Kit mutations in TNBC were hardly encountered in DNA databases. Our observations agree with a former analysis that found a single activating mutation in the c-Kit gene in only one patient of a cohort of 171 TNBC cases [27]. Collectively, these data indicate that in breast cancer the expression of wild-type c-Kit is predominantly observed in TNBC.

We observed that c-Kit function in TNBC cells depended on SCF-binding to be activated. Upon activation, c-Kit induces cell proliferation, survival, and migration. Here, neither cell proliferation nor cell survival was stimulated by SCF. Considering that responses that are mediated by the SCF/c-Kit axis are tissue-specific [14], we speculate that c-Kit activation might be irrelevant for the proliferation and survival of TNBC cells. Accordingly, we and others [28] found that the expression of c-Kit did not correlate with TNBC patient mortality, suggesting that c-Kit might not have an impact on tumor cells behavior. Unlike proliferation and survival, c-Kit activation induced an increment in the migratory potential of two cell lines. SCF/c-Kit axis regulates the migration of colorectal cancer [31] and cervical cancer [20], but to our knowledge this is the first report showing that c-Kit mediates TNBC cell migration. It has been demonstrated that the PI3K/Akt [32] and JAK2/STAT3 [33] cascades are important regulators of TNBC cell migration. Correspondingly, we observed that SCF stimulation induced the activation of Akt and STAT3 in migrating cells. However, in order to demonstrate that c-Kit-induced migration depends on the activation of those pathways, inhibition of key elements of the cascades would be necessary.

The migration of tumor cells is a driving mechanism of breast cancer invasion and metastasis. Our results indicated that c-Kit induces cell migration in vitro. However, we did not find an association of c-Kit expression and reduced metastasis-free survival in human populations. Since c-Kit activity depends on the presence of SCF, we consider that a likely explanation might be a deficiency of SCF. This is supported by the fact that analysis of expression datasets demonstrated the absence of SCF overexpression in TNBC patients. In addition, it is accepted that c-Kit oncogenic potential relies on the presence of gain of function mutations. Interestingly, c-Kit mutations are not reported in TNBC according to DNA sequencing datasets, suggesting that the receptor might not play a pro-metastatic role in this type of cancer. Accordingly, analysis of a cohort of breast cancer patients showed that the expression of c-Kit was significantly associated with TNBC subtypes, but the presence of positive lymph nodes and shorter disease-free survival periods were exclusively associated with the expression of Ki67 [34].

c-Kit is associated with the induction of various signaling pathways. In this work we observed that STAT3 was efficiently activated by SCF in all cell lines that were tested. STAT3 is a key regulator and promoter of breast cancer [35]. The importance of active STAT3 in TNBC has been recently unveiled by Nakagawa and colleagues [36], who reported that the nuclear expression of pSTAT3 was significantly associated with short relapse-free survival and a poor prognosis in TNBC patients, but not in ER(+) breast cancer women. The JAK2/STAT3 cascade can be activated by a variety of receptors, including RTKs. The relevance of c-Kit-mediated activation of STAT3 was demonstrated in a Phase II study on the c-Kit inhibitor Nilotinib in melanoma patients [11]. The authors found that positive clinical responses to Nilotinib were associated with a reduction of STAT3 phosphorylation in tumors during follow up. They also demonstrated that Nilotinib significantly decreased STAT3 phosphorylation in melanoma cell lines and inhibited cell proliferation as efficiently as specific STAT3 inhibitors [11], suggesting that the interaction of c-Kit with STAT3 should be considered when assessing responses to c-Kit inhibitors in cancer models.

Several RTK-inhibitor drugs can block c-Kit enzymatic activity. Here, we tested the cytotoxic effect of five FDA-approved TKIs targeting c-Kit. Dissimilar responses were obtained. Imatinib targets Abl and c-Kit. Although a former study found that Abl is overexpressed in a subgroup of TNBC and suggested Imatinib as a potential treatment [37], we observed that Imatinib had the weakest cytotoxic activity among all TKIs that were tested, and that it was not superior to Doxorubicin. A likely explanation might be that Imatinib has shown a potent cytotoxic effect against tumor cells having mutated, constitutively active c-Kit, and TNBC cells that were assessed here had non-mutated c-Kit. On the other hand, we observed a modest cytotoxic effect of Dasatinib, Sorafenib, and Sunitinib, that was weaker than the effect of Doxorubicin. Accordingly, the anticancer activity of Dasatinib [38] and Sunitinib in cell lines [39], and Sorafenib in human trials [40] has demonstrated to be moderate. Interestingly, we observed that TNBC cells were sensitive to Lapatinib, showing a cytotoxic effect that was similar to Doxorubicin. Lapatinib is indicated for the treatment of HER2-positive breast cancer patients. It targets not only HER2 but also EGFR, so we hypothesize that the cytotoxicity that was observed was mediated by the inhibition of EGFR. Our analysis of data from the METABRIC cohort demonstrated the expression of EGFR in 40% of basal subtype cancer patients; most of them did not co-express HER2. Additionally, the association of EGFR expression with TNBC has been previously reported in a human population [29]; thus, to confirm our hypothesis, the expression and function of EGFR in TNBC cell lines, as well as the inhibitory effect of Lapatinib are currently being explored.

In contrast with the aforementioned TKIs, Nilotinib displayed a consistent cytotoxic effect on all cell lines, with lower IC50 values than Doxorubicin. Nilotinib, a small-molecule drug that inhibits c-Kit and Abl kinases, is indicated for the treatment of Imatinib-resistant chronic myelogenous leukemia. Here, we observed the expression of c-Kit and Abl in TNBC cells, suggesting that the presence of both kinases may contribute to the cytotoxic effect of Nilotinib. The effect of Nilotinib on breast cancer has been barely investigated. In former studies, Nilotinib was demonstrated to reduce extracellular matrix degradation and invasion [41], and to induce apoptosis in TNBC cell lines [42]. In a recently published study using integrated bioinformatics approaches, a set of differentially expressed genes were found in breast cancer. The authors further analyzed the products of the identified genes by molecular docking stimulation and detected that Nilotinib was a top ranked candidate drug for the treatment of breast tumors [43]. Although this study seems to support our findings, a proper comparison is not possible because the authors did not categorize breast tumors by subtype. Nevertheless, taken together these results reinforce the proposal of Nilotinib as a potential drug for TNBC, and justify deeper studies to unveil the mechanisms, specificity, and side effects of Nilotinib cytotoxicity, both in TNBC cell lines and animal models.

## 4. Materials and Methods

### 4.1. Cell Lines

TNBC cell lines HCC-1806 (ATCC, CRL 2335™), HCC-1937 (ATCC, CRL-2336™), HCC-70 (ATCC, CRL-2315™), and MDA-MB-468 (ATCC, HTB-132™), and the chronic myelogenous leukemia-derived K562 cell line (ATCC, CCL-243™) were purchased from the American Type Culture Collection (ATCC, Rockville, MD, USA). The cell lines were authenticated by DNA profiling using short tandem repeat (STR) analysis on an AmpFlSTR^®^ Identifier™ PCR Amplification System at the National Institute of Genomic Medicine (INMEGEN), Mexico City, Mexico. The cells were used between passage 3 and passage 15. All the cells were maintained in Roswell Park Memorial Institute (RPMI)-1640 medium (Biowest SAS, Nuaillé, France) that was supplemented with 10% heat-inactivated Fetal Bovine Serum (FBS) (Biowest SAS, Nuaillé, France), 100 U/mL penicillin, and 100 mg/mL streptomycin (Invitrogen, Carlsbad, CA, USA). The cells were maintained in a 5% CO_2_ humidified atmosphere at 37 °C.

### 4.2. Reverse Transcriptase-PCR (RT-PCR)

Total RNA was isolated from the cells using the RNeasy kit (Qiagen GmbH, Hilden, Germany), according to the instructions of the manufacturers. cDNA was synthesized using the Maxima First Strand cDNA Synthesis Kit for RT-qPCR (Thermo Scientific Co., Waltham, MA, USA) following the manufacturer’s instructions. A total of 250 ng cDNA was amplified using the qTaq PCR Core Kit (qARTA Bio Inc., Carson, CA, USA). The oligonucleotides that were used for gene expression analysis are shown in Table 2. Reverse transcription was carried out at 25 °C for 10 min, followed by 50 °C for 15 min, and 85 °C for 5 min. The PCR protocol was: 95 °C for 30 s, followed by 40 cycles of 94 °C for 30 s, the annealing temperature for each gene (Table 2) for 30 s, followed by 72 °C for 30 s. Amplification of GAPDH was included as an internal control.

### 4.3. Analysis of Protein by Western Blot

The total protein was extracted using lysis buffer (50 mN Tris-HCl, pH 7.4; 150 mM NaCl; 1 mM EDTA; 1% NP40; 0.25% sodium deoxycholate) containing 10 µL/mL phosphatase inhibitors (Sigma-Aldrich, St. Louis, MO, USA), and 100 µL/mL complete protease inhibitor cocktail (Roche Diagnostics GmbH.; Manheim, Germany). The protein concentration was measured using the DC protein assay kit (Bio-Rad Laboratories, Inc.; Hercules, CA, USA). A total of 30 µg/lane of cell protein was resolved by 10% SDS-PAGE, and transferred onto polyvinylidene fluoride (PVDF) membranes (EMD Millipore, Billerica, MA, USA). PVDF membranes were incubated at 4 °C overnight with the following primary antibodies: c-Kit (Cat. sc13508; Santa Cruz Biotechnology Inc., Santa Cruz, CA, USA), and phospho-c-Kit (Cat. 48347; Cell Signaling Technology, Inc., Danvers, MA, USA) diluted 1:500; Akt (Cat. sc8312; Santa Cruz Biotechnology Inc.), phospho-Akt (Cat. sc7985R; Santa Cruz Biotechnology Inc.), ERK1/2 (Cat. GTX17942; GeneTex, Inc., Irvine, CA, USA), phospho-ERK1/2 (Cat. sc7383; Santa Cruz Biotechnology Inc.), STAT3 (Cat. GTX108630; GeneTex, Inc.), phospho-STAT3 (Cat. sc8059; Santa Cruz Biotechnology Inc.), and GAPDH (Cat. 100118; GeneTex, Inc.) that was diluted 1:1000. Horseradish peroxidase-conjugated secondary anti-rabbit IgG (Cat. GTX213111-01; GeneTex, Inc.), and anti-mouse IgG (Cat. GTX213110-01; GeneTex, Inc.) were used diluted 1:10,000. The detection of specific proteins was carried out by chemiluminescence using the SuperSignal WestFemto Maximum Sensitivity Substrate (Thermo Scientific, Waltham, MA, USA). Densitometry analysis was performed using the ImageJ 1.8.0_172 (NIH, Bethesda, MD, USA).

### 4.4. Flow Cytometry

Cell membrane c-Kit was measured by flow cytometry. A total of 1 × 10^6^ cells were washed with sterile phosphate buffered saline (PBS) and fixed with 0.1% paraformaldehyde containing 0.1% sodium azide (in PBS, pH7.4) for 15 min at 37 °C. After washing two times with PBS, the cells were incubated with the phycoerythrin-conjugated mouse monoclonal anti-human c-Kit/CD117 (R&D Systems, Minneapolis, MN, USA) and diluted 1:4 in PBS for 1 h at room temperature. After washing two times with PBS, the cells were resuspended in PBS for flow cytometry analysis. The cells that were exposed to the same experimental conditions, except for incubation with anti-c-Kit antibody, were included as unstained control cells. The cells were analyzed using a FACS Calibur flow cytometer (BD Biosciences, San Jose, CA, USA). A total of 10,000 events per sample were captured. To identify cells expressing membrane c-Kit, single parameter histograms were generated. The percentage of positive cells was analyzed using the BD FACStation 6.1 software (BD Biosciencies, San Jose, CA, USA).

### 4.5. c-Kit DNA Sequencing

DNA was isolated from the TNBC cell lines HCC-1806, HCC-1937, HCC-70, and MDA-MB-468. Sanger sequencing was completed for c-Kit exons 9, 10, 11, 13, and 17 by PCR amplification. The PCR products were purified using the Wizard SV Gel and PCR Clean-up System (Promega Co., Madison, WI, USA). The purified fragments were sequenced with the following forward and reverse oligonucleotides:

Exon 9: Forward TAGAGTAAGCCAGGG/Reverse AATCATGACTGATATGGT

Exon 10: Forward GATCCCATCCTGCCAAAGTT/Reverse ATTGTCTCAGTCATTAGAGCAC

Exon 11: Forward CAGGTAACCATTTATTTGT/Reverse TCATTGTTTC AGGTGGAAC

Exon 13: Forward ATCAGTTTGCCAGTTGTGCT/Reverse TTTATAATCT AGCATTGCC

Exon 17: Forward TACAAGTTAAAATGAATTTAAATGGT/Reverse AAGTGAAACTAAAAATCCTTTGC

The sequencing was performed on an Applied Biosystems 3730 Genetic Analyzer (Applied Biosystems, Waltham, MA, USA). The sequences that were obtained were analyzed using the NCBI (National Center for Biotechnology Information) Basic Local Alignment Search Tool (BLAST) (https://blast.ncbi.nlm.nih.gov/Blast.cgi, accessed on 15 November 2018) [50].

### 4.6. Proliferation Assay

The cells were incubated with increasing concentrations (1, 5, 10, 25, 50, or 100 ng/mL) Stem Cell Factor (SCF) (PeproTech Ltd., Cranbury, NJ, USA) or left untreated. The number of viable cells was evaluated at 24, 48, 72, and 96 h, using the colorimetric 3-(4,5-dimethylthiazol-2-yl)-2,5-diphenyltetra-zolium bromide (MTT) assay.

### 4.7. Migration Assay

The Boyden Chamber migration assay was used to evaluate the effect of SCF on cell migration. The cells were starved by incubating for 4 h in FBS-free RPMI medium. To inhibit c-Kit activity, the cells were incubated with 10 µM ISCK03 inhibitor (4-t-Butylphenyl-N-(4-imidazol-1-yl-phenyl) sulfonamide) (Santa Cruz Biotechnology Inc., Santa Cruz, CA, USA) during the last 2 h of starving. A total of 2.5 × 10^5^ cells were washed three times with chelating buffer (5% bovine serum albumin, 1 mM MgSO_4_-7H_2_O in FBS-free RPMI) and then seeded onto cell culture inserts (ThermoScientific™ Nunc™, Cell Culture Inserts in Carrier Plate System, 8 µM pore, 24-well format) (ThermoScientific Co., Waltham, MA, USA) in 300 µL FBS-free RPMI. A toal of 500 µL FBS-free RPMI containing 1, 10, and 100 ng/mL SCF, RPMI that was supplemented with 10% FBS as positive control, and FBS-free RPMI as a negative control were loaded into the lower chambers. The plates were incubated for 24 h. The inserts were washed with PBS, and the migrating cells were fixed with glutaraldehyde (1.1% in PBS) for 20 min at room temperature. The cells were stained with a crystal violet solution (0.5% crystal violet; 20% methanol) for 5 min. The inserts were air-dried and then incubated in 10% acetic acid for 15 min. The optical density was measured at 570 nm in a plate reader ELx800 (Bio Tek Instruments, Winooski, VT, USA).

### 4.8. Cytotoxicity Assays

The concentration of Doxorubicin, Imatinib, Dasatinib, Lapatinib, Sorafenib, Sunitinib, and Nilotinib (all from Merck KGaA, Darmstadt, Germany) that was needed to inhibit cell growth by half (inhibitory concentration (IC50)) was calculated by incubating HCC-1806, HCC-1937, HCC-70, and MDA-MB-468 cells with increasing concentrations of each drug for 48 h. The cell viability was measured using the colorimetric MTT assay. To evaluate the effect of c-Kit activation on cell survival, the cells were pre-incubated with 100 ng/mL SCF for 6 h. The cells were subsequently treated with the calculated IC50 of Doxorubicin for 48 h. The cell viability was evaluated by the MTT assay.

### 4.9. Publicly Available Data Sets Analysis

The METABRIC (Molecular Taxonomy of Breast Cancer International Consortium), containing DNA and RNA sequencing data from 299 primary TNBC samples [15,16] dataset were downloaded from the cBioPortal database (http://www.cbioportal.org/, accesed on 1 June 2022). Standardized survival data of the METABRIC cohort were downloaded from the cBioPortal database. Transcriptome data from TCGA (The Cancer Genome Atlas project) [17], containing data from 168 TNBC tumors, were downloaded using GDC (Genomic Data Commons data portal) RNA tools (https://portal.gdc.cancer.gov/, accessed on 4 May 2022). The Gene expression-based Outcome for Breast cancer Online (GOBO) tool (http://co.bmc.lu.se/gobo, accessed on 7 February 2020) [51,52] was used to evaluate the level of expression of target RTKs and intracellular tyrosine kinases in TNBC patients. In the aforementioned datasets, tumor samples are categorized according to the PAM50 method into five intrinsic subtypes (basal-like, luminal A, luminal B, HER2-enriched, and normal-like) [53]; thus, for our analysis we considered patients of the basal-like subtype because they lack overexpression of ER/PR and HER2 and have been demonstrated to share most clinical and genetic characteristics with TNBC tumors [54].

### 4.10. Statistical Analysis

Statistical analyses were performed using the GraphPad Prism 8.02 Software (GraphPad Software Inc., San Diego, CA, USA). All the data are presented as the mean ± standard error of the mean. Differences between the treatments were evaluated using a one-way analysis of variance (ANOVA) test and a Dunnett post hoc test. 95% confidence intervals were calculated, and *p* values < 0.05 were considered statistically significant.

## 5. Conclusions

In this work, the expression of functional, non-mutated c-Kit in TNBC was demonstrated. The activation of c-Kit promoted cell migration that was associated with the activity of the MAPKs and STAT3 signaling cascades. Although c-Kit activation by its ligand SCF induced the phosphorylation of STAT3, which is linked with cell survival, we did not observe changes in the response to Doxorubicin in cells that were pre-incubated with SCF. These observations are consistent with the fact that the expression of c-Kit in tumor samples had no effect on patient´s survival. c-Kit is a target molecule for cancer treatment using TKIs. Our results showed that TNBC cells are indeed susceptible to the cytotoxic effect of TKIs targeting c-Kit. However, it is worthy of note that only Nilotinib showed a significantly higher cytotoxic effect than Doxorubicin, which is the treatment of choice for women with TNBC. Comparative analysis of the expression of the different tyrosine kinases that are targeted by the TKIs included in this work showed that c-Kit is more frequently expressed than the majority of targeted molecules in TNBC tumor samples; thus, c-Kit might be considered a potential marker for the treatment of TNBC using TKIs, in particular Nilotinib.

## Figures and Tables

**Figure 1 ijms-23-08702-f001:**
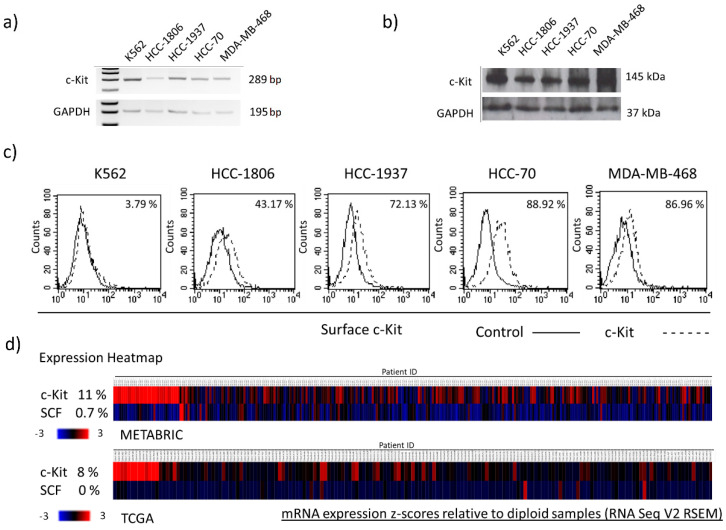
**Expression of c-Kit in TNBC cell lines and tumors.** The expression of c-Kit was evaluated in HCC-1806, HCC-1937, HCC-70, and MDA-MB-468 TNBC-derived cell lines by (**a**) RT-PCR, and (**b**) Western blot; (**c**) the presence of c-Kit at the cell membrane was evaluated by flow cytometry, the proportion of cells showing surface c-Kit is indicated. K562 cells were included as a c-Kit-expression positive control, and the expression of GAPDH was used as internal control for RT-PCR and Western blot assays. (**d**) The expression of c-Kit and its ligand SCF was investigated in tumor samples using transcriptomic data from the METABRIC (N = 299 samples), and TCGA (N = 168 samples) datasets. Gene overexpression was considered for samples showing a z-score relative to diploid samples > 2 (EXP > 2). Gene underexpression was considered when the z-score relative to diploid samples was <2 (EXP < 2).

**Figure 2 ijms-23-08702-f002:**
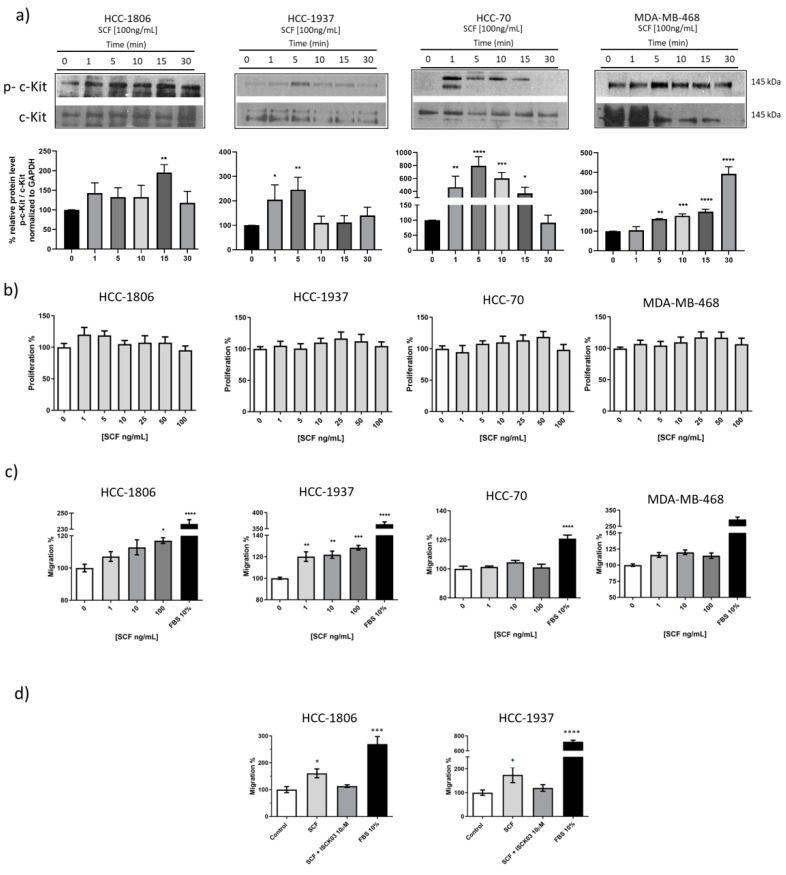
**Effect of c-Kit activation on TNBC cell proliferation and migration.** (**a**) Time-course of c-Kit activation by SCF. HCC-1806, HCC-1937, HCC-70, and MDA-MB-468 cells were incubated with 100 ng/mL SCF. Phosphorylation of the receptor was evaluated by Western blotting at the indicated time points. Densitometric analysis indicated as percentage is shown below each blot. (**b**) For the proliferation assay, the cells were treated with increasing concentrations of SCF. Cell proliferation was evaluated at 48 h using the MTT colorimetric assay. (**c**) The migration of cells was evaluated using the Boyden chamber assay using 1, 10, and 100 ng/mL SCF, 10% FBS as positive control, or FBS-free medium as negative control that were loaded into the lower chambers. Cell migration was evaluated after 24 h. (**d**) To inhibit c-Kit phosphorylation, HCC-1806 and HCC-1937 cells were pre-incubated with 10 µM ISCK03. Cell migration was then evaluated by Boyden chamber assays using FBS-free medium (Control), 100 ng/mL SCF (SCF), 100 ng/mL SCF, and the indicated concentration of ISCK03 (SCF + ISCK03 10 µM), or 10% FBS. Values represent the average of three independent experiments. Error bars indicate the standard error of the mean. * *p* < 0.05, ** *p* < 0.005, *** *p* < 0.0005, and **** *p* < 0.0001 vs. negative control values (Dunnett test).

**Figure 3 ijms-23-08702-f003:**
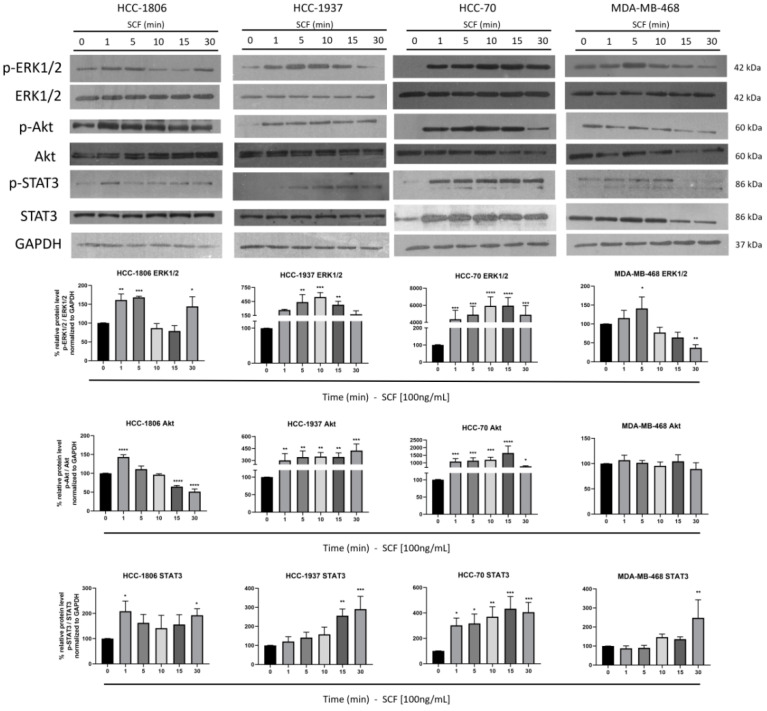
**The effect of c-Kit activation on cell signaling cascades.** HCC-1806, HCC-1937, HCC-70, and MDA-MB-468 cells were incubated with 100 ng/mL SCF for the indicated time points. Protein lysates were analyzed by 10% SDS-PAGE. Phosphorylation of clue proteins was detected using specific antibodies to p-ERK1/2, ERK1/2, p-Akt, Akt, p-STAT3, and STAT3. The detection of GAPDH was included as a control. Representative blots from triplicate experiments are shown. Densitometric analysis, indicated as percentage, is shown below the blots. The values represent the average of three independent experiments. Error bars indicate the standard error of the mean. * *p* < 0.05, ** *p* < 0.005, *** *p* < 0.0005, and **** *p* < 0.0001 vs. values that were obtained at time 0 (Dunnett test).

**Figure 4 ijms-23-08702-f004:**
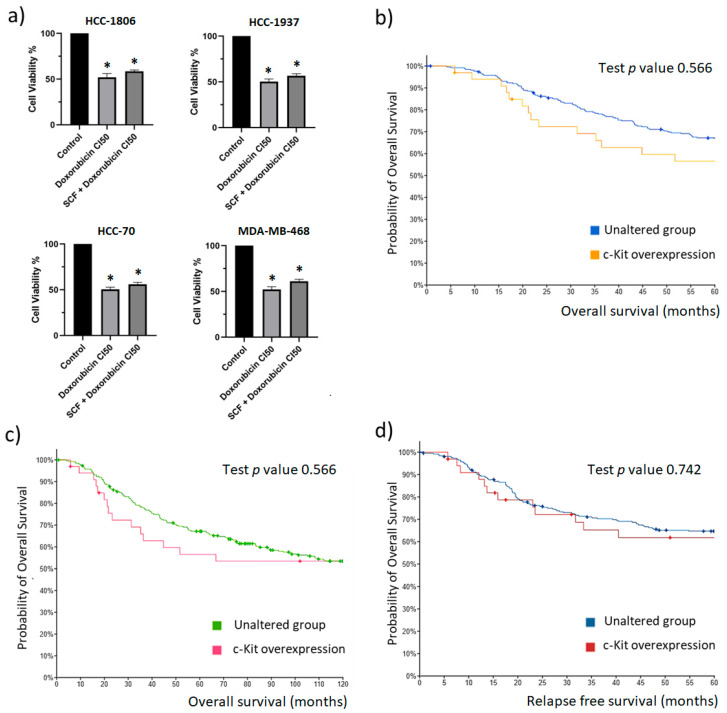
**Effect of c-Kit activation on the cytotoxic activity of Doxorubicin.** (**a**) HCC-1806, HCC-1937, HCC-70, and MDA-MB-468 cells were pre-incubated with 100 ng/mL SCF and then treated with the calculated IC50 of Doxorubicin. Cell viability was measured after 48 h by the colorimetric MTT assay. The values represent the average of three independent experiments. The error bars indicate the standard error of the mean. * *p* < 0.05 vs. the untreated control values (Dunnett test). (**b**) The association of tumor c-Kit overexpression with 5-year overall survival of TNBC patients was explored using METABRIC datasets. (**c**) The association of c-Kit overexpression with 10-years overall survival. (**d**) The association of c-Kit overexpression with the level of 5-year relapse-free survival. Statistical comparison of samples with c-Kit overexpression and tumors with unaltered expression of c-Kit is presented. Standardized survival data of the METABRIC cohort (unaltered c-Kit expression N = 264; c-Kit overexpression N = 33) were downloaded from the cBioPortal database.

**Figure 5 ijms-23-08702-f005:**
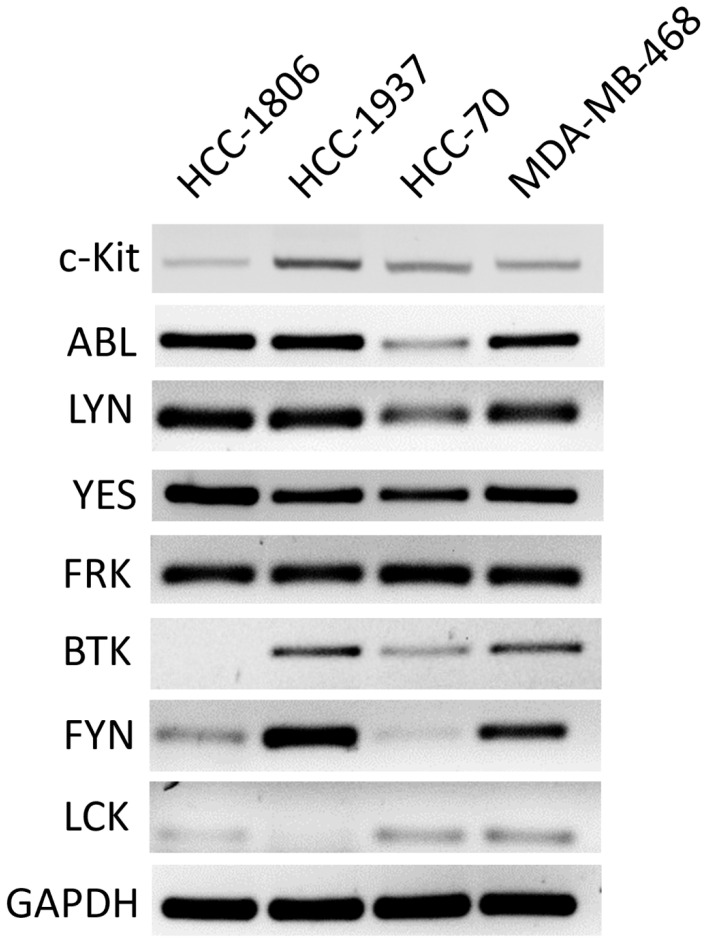
**Expression of Receptor Tyrosine Kinases (RTKs) and intracellular tyrosine kinases in TNBC cell lines.** Total RNA was isolated from HCC-1806, HCC-1937, HCC-70, and MDA-MB-468 cells. The expression of c-Kit, ABL, LYN, YES, FRK, BTK, FYN, and LCK was evaluated by RT-PCR. The expression of GAPDH was included as a control. Amplified fragments were analyzed by electrophoresis in agarose gels. Representative gels from triplicate experiments are presented.

**Figure 6 ijms-23-08702-f006:**
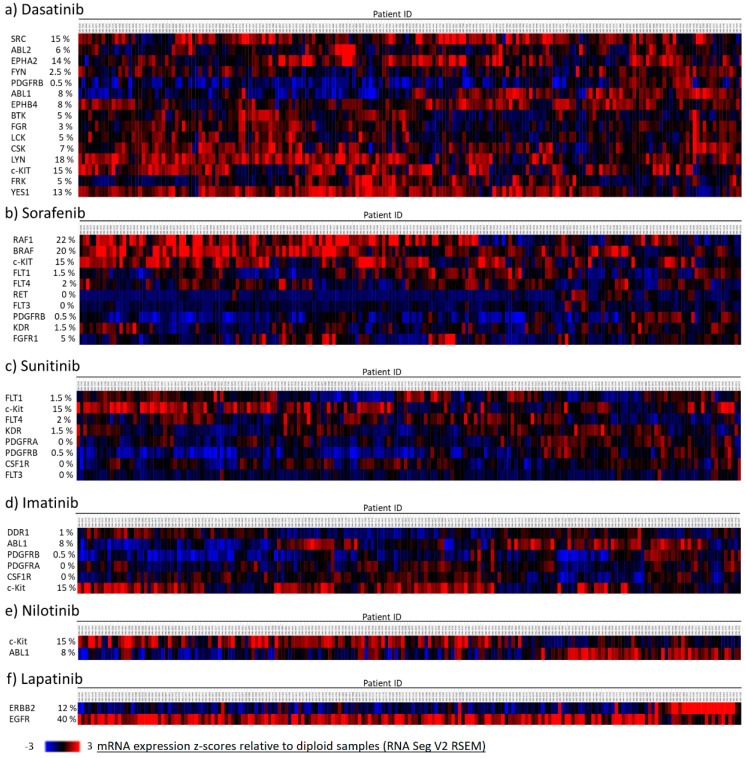
**Expression of TKI-targeted RTKs and intracellular tyrosine kinases in TNBC tumor samples**. The expression of the RTKs and intracellular tyrosine kinases that are targeted by (**a**) Dasatinib, (**b**) Sorafenib, (**c**) Sunitinib, (**d**) Imatinib, (**e**) Nilotinib, and (**f**) Lapatinib was investigated using the METABRIC transcriptomic datasets. A cohort of 199 patients was analyzed. Samples with a z-score relative to diploid samples >2 were considered to have overexpression of the specific target molecule. The percentage of patients showing overexpression of each tyrosine kinase is presented.

**Table 1 ijms-23-08702-t001:** Calculated IC50 values for Tyrosine Kinase Inhibitors.

Tyrosine Kinase Inhibitor	HCC-1806 IC50 (nM) ± SD	HCC-1937 IC50 (nM) ± SD	HCC-70 IC50 (nM) ± SD	MDA-MB-468 IC50 (nM) ± SD
Imatinib	51,800 ± 5853	24,750 ± 1819	24,800 ± 1042	24,310 ± 905.1
Dasatinib	340 ± 30.8	3120 ± 153.7	10,230 ± 1198	9210 ± 226.9
Sorafenib	250 ± 13.6	585 ± 47.3	13,660 ± 1119	130 ± 11.3
Sunitinib	3640 ± 550	2910 ± 190.6	3420 ± 277.6	285 ± 13.3
Nilotinib	46.0 ± 4.3	11.0 ± 1.6	31.0 ± 2.9	23.0 ± 4.1
Lapatinib	26.0 ± 5.8	739.0 ± 35.3	80.0 ± 8.9	1380 ± 125.4
Doxorubicin	180.5 ± 19.6	191.2 ± 17.6	700.0 ± 180	127.7 ± 8.0

**Table 2 ijms-23-08702-t002:** Oligonucleotides that were used for the analysis of tyrosine kinases gene expression by RT-PCR.

Tyrosine Kinase	Forward 5′-3′	Reverse 5′-3′	Annealing Temperature (°C)	Reference
c-Kit	GTTGAGGCAACTTGCTTATGG	GCTTCTGCATGATCTTCCTG	58	[44]
LCK	CGATCTGGTCCGCCATTACA	ATGGTTTCTGGGGCTTCTGG	60	[45]
YES	GCGGTAGCAGCGACTCAT	TGCACCTCCAAAAGGCGTTA	59	*
FYN	AAAACTGACGGAGGAGAGGG	TTCACACCTCCAAAGACGGT	61	[46]
FRK	CGAGCAGGTGACAAACTTCA	TGCCTGTAGGCTTCTGTCCT	61	[47]
LYN	TGAAGACTCAACCAGTTCCAGA	GGTGGATGCCATCATAGGGG	60	*
BTK	GAGAAGCTGGTGCAGTTGTA	GGCCGAAATCAGATACTTTAAC	51.9	[48]
ABL	AGGTAGCTGAGGAGCTTGGGAGAG	TTTGCTTTCGAGGCAGTGCTGGGG	66	[49]
GAPDH	CAGCCTCAAGATCATCAGC	ATGATGTTCTGGAGAGCCC	50	[46]

* Oligonucleotide sequences that were designed by the authors using data from the NCBI (National Center for Biotechnology Information).

## Data Availability

Not applicable.

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
