# Peer review of "c-Kit Induces Migration of Triple-Negative Breast Cancer Cells and Is a Promising Target for Tyrosine Kinase Inhibitor Treatment"

_ijms, 2022, doi:10.3390/ijms23158702_

Round 1

Reviewer 1 Report

Interesting approach in this population with limited treatment options . Additional modalities and new therapeutical targets are urgently needed. I hope the results can translate in clinical benefit for TNBC patients

Author Response

14-July-2022

Response to REVIEWER 1.

Please find enclosed the response to your comments and suggestions to the manuscript entitled “c-Kit Induces Migration of Triple-Negative Breast Cancer Cells and is a Promising Target for Tyrosine Kinase Inhibitor Treatment” by José A. López-Mejía, Luis F. Tallabs-Utrilla, Pablo Salazar-Sojo, Jessica C. Mantilla-Ollarves, Manuel A. Sánchez-Carballido, and Leticia Rocha-Zavaleta, to be submitted to the International Journal of Molecular Sciences, in the Special Issue “State-of-the-Art Molecular Oncology in Mexico”.

Comments and Suggestions for Authors:

Interesting approach in this population with limited treatment options. Additional modalities and new therapeutical targets are urgently needed. I hope the results can translate in clinical benefit for TNBC patients

AUTHORS: We do thank the reviewer for the comments. Your considerations are very valuable for us. You can be sure that we will work hard to get enough preclinical evidence to sustain future clinical trials.

Yours Sincerely

Leticia Rocha-Zavaleta, PhD

Reviewer 2 Report

The article is devoted to the study of the expression and effects of the c-kit proto-oncogene, which encodes a transmembrane tyrosine kinase receptor on cell lines of triple-negative breast cancer. The manuscript presents important data that are of fundamental significance. In addition, the article provides evidence of the feasibility of using c-kit as a potential therapeutic target for triple-negative cancer and suggests the most effective drugs from the list of tyrosine kinase inhibitors. Comment 1. Interesting results regarding c-Kit activation and the enhanced cell migration ability associated with the activity of MAPK and STAT3 signaling cascades were obtained.  However, the study did not reveal an association between the level of c-Kit expression and overall and disease-free survival rates. In my opinion, it would be helpful to discuss the relationship between the marker and long-term metastatic-free survival rates (for example, taking into account available databases or literature reviews). Comment 2. In conclusion, it would be useful to present a general scheme for the involvement of c-Kit as a target for anticancer treatment in triple-negative breast cancer.Comment 3. In the Materials and Methods section, I recommend to present the gating strategy of c-kit+ cells in flow cytometry. Comment 4. I wonder if additional cell passages were performed. If so, how many passages were performed before the study and how was the purity of cell lines assessed? Comment 5. Fig 3 (HCC1937 – p-STAT3). The results of Western blot analysis of this block are not of good quality. If possible, replace it with a higher quality image. In addition, there are minor spelling mistakes and typos in the article (for example, line 481). Check the manuscript again for typos and spelling errors.

In general, highly appreciating the quality of the manuscript, I recommend it for publication in the International Journal of Molecular Sciences with minor revisions.

Author Response

14-July-2022

Response to RVIEWER 2.

Please find enclosed the response to your comments and suggestions to the manuscript entitled “c-Kit Induces Migration of Triple-Negative Breast Cancer Cells and is a Promising Target for Tyrosine Kinase Inhibitor Treatment” by José A. López-Mejía, Luis F. Tallabs-Utrilla, Pablo Salazar-Sojo, Jessica C. Mantilla-Ollarves, Manuel A. Sánchez-Carballido, and Leticia Rocha-Zavaleta, to be submitted to the International Journal of Molecular Sciences, in the Special Issue “State-of-the-Art Molecular Oncology in Mexico”.

REVIEWER 2.

Comments and Suggestions for Authors

The article is devoted to the study of the expression and effects of the c-kit proto-oncogene, which encodes a transmembrane tyrosine kinase receptor on cell lines of triple-negative breast cancer. The manuscript presents important data that are of fundamental significance. In addition, the article provides evidence of the feasibility of using c-kit as a potential therapeutic target for triple-negative cancer and suggests the most effective drugs from the list of tyrosine kinase inhibitors.

 Comment 1. Interesting results regarding c-Kit activation and the enhanced cell migration ability associated with the activity of MAPK and STAT3 signaling cascades were obtained.  However, the study did not reveal an association between the level of c-Kit expression and overall and disease-free survival rates. In my opinion, it would be helpful to discuss the relationship between the marker and long-term metastatic-free survival rates (for example, taking into account available databases or literature reviews).  

AUTHORS: We do thank the reviewer for this important comment. We have included the following paragraph in the Discussion section along with a new reference:

“Migration of tumor cells is a driving mechanism of breast cancer invasion and metastasis. Our results indicated that c-Kit induces cell migration in vitro; however, we did not find an association of c-Kit expression and reduced metastasis-free survival in human populations. Since c-Kit activity depends on the presence of SCF, we consider that a likely explanation might be a deficiency of SCF, this is supported by the fact that analysis of expression datasets demonstrated the absence of SCF overexpression in TNBC patients. In addition, it is accepted that c-Kit oncogenic potential relies on the presence of gain of function mutations. Interestingly, c-Kit mutations are not reported in TNBC according to DNA sequencing datasets, suggesting that the receptor might not play a pro-metastatic role in this type of cancer. Accordingly, analysis of a cohort of breast cancer patients showed that the expression of c-Kit was significantly associated with TNBC subtype, but the presence of positive lymph nodes and shorter disease-free survival periods were exclusively associated with the expression of Ki67 [34]”

[34] Constantinou, C.; Papadopoulos, S.; Karyda, E.; Alexopoulos, A.; Agnati, N.; Batistatou, A.; Harisis, H. Expression and clinical significance of claudin-7, PDL-1, PTEN, c-Kit, c-Met, c-Myc, ALK, CK5/6, CK17, p53, EGFR, Ki67, p63 in triple-negative breast cancer- a single centre prospective observational study.  2018, 32, 303-311. doi: 10.21873/in vivo.11238.

Comment 2. In conclusion, it would be useful to present a general scheme for the involvement of c-Kit as a target for anticancer treatment in triple-negative breast cancer.

AUTHORS: We completely agree with the usefulness of a general scheme, in fact we have provided the following Graphical Abstract that shows what the reviewer suggests (Please see the attachment)

Comment 3. In the Materials and Methods section, I recommend to present the gating strategy of c-kit+ cells in flow cytometry. 

AUTHORS: We thank the reviewer for this comment. As a direct immunofluorescence for detection of a single marker was conducted, we generated Single Parameter Histograms for control and stained cells, they were then compared and analyzed. The following text describing the procedure has been included in the Materials and Methods section, subsection 4.4. Flow Cytometry:

 “Cells exposed to the same experimental conditions, except for incubation with anti-c-Kit antibody were included as unstained control cells. Cells were analyzed using a FACS Calibur flow cytometer (BD Biosciences, CA, USA). A total of 10,000 events per sample were captured. To identify cells expressing membrane c-Kit, single parameter histograms were generated. The percentage of positive cells was analyzed using the BD FacStation software.”  

Comment 4. I wonder if additional cell passages were performed. If so, how many passages were performed before the study and how was the purity of cell lines assessed?

AUTHORS: This in an important methodological question, we do thank the reviewer for asking. The cells were obtained from ATCC especially for this project, their purity was demonstrated by authentication using DNA profiling techniques, and they were used up to passage 15. We have included this information in the Materials an Methods section, subsection  4.1. Cell lines as follows:

Cell lines were authenticated by DNA profiling using short tandem repeat (STR) analysis on an AmpFlSTR® Identifier™ PCR Amplification System at the National Institute of Genomic Medicine (INMEGEN), Mexico City.  Cells were used between passage 3 and passage 15.”

 Comment 5. Fig 3 (HCC1937 – p-STAT3). The results of Western blot analysis of this block are not of good quality. If possible, replace it with a higher quality image.

AUTHORS: The reviewer is right. We have changed the Western blot image for one with higher quality.   

In addition, there are minor spelling mistakes and typos in the article (for example, line 481). Check the manuscript again for typos and spelling errors.

AUTHORS: We apologize for spelling mistakes, the manuscript has been reviewed and corrected by a professional English editing service.

Yours Sincerely

Leticia Rocha-Zavaleta, PhD

Reviewer 3 Report

Summary:

An overall good paper about the potential role of c-Kit as a potential marker for TNBC treatment. The author did a good job presenting their work and science. They emphasize on the role of c-Kit on cell migration, and the effect of its activation through SCF on downstream effectors. The author drew the right conclusions from the experiments without hyperbolizing his results. With that said, I have a few comments regarding some of the experiments.

Comments: 

1- Figure 1: k562 cells were used as positive control for c-kit expression, however no expression can be seen in the WB.

2- Figure 2-A: The quality of WB for HCC-1937 is poorcould be improved

3- I suggest adding WB quantification with statistics to further highlight the results (figures 1, 2, and 3)

There are a few spelling mistakes across the paper, quick review of grammar.
